# A Study on Bandwagon Consumption Behavior Based on Fear of Missing Out and Product Characteristics

**Inwon Kang and Ilhwan Ma *** 

Department of International Business and Trade, Kyung Hee University, Seoul 02447, Korea; iwkang@khu.ac.kr
* Correspondence: ma.ilhwan@khu.ac.kr

**Abstract:** There have been unusual collective consumption phenomena that consumer behavior conforms to as part of the pursuit of specific brands (e.g., teenagers wearing the same brand jacket). In order to explain bandwagon consumption behavior, previous studies focused on brand, personal traits, and group characteristics. However, previous studies seem somewhat limited in explaining excessive bandwagon consumption. This study addresses a psychological trait, the Fear of Missing out (FoMO), which describes why people want to belong to a main group. Along with FoMO, product characteristics influence bandwagon consumption behavior because consumers may show different behaviors for different product categories. Hence, this study combines FoMO levels (high/low) and product category (luxury/necessity) to explain bandwagon consumption behavior. The results reveal that the combination of high FoMO and luxury shows the strongest bandwagon consumption behavior. The next strongest combination is low FoMO and necessity, followed by high FoMO and necessity, and low FoMO and luxury. Based on these findings, this study might present implications for understanding excessive bandwagon consumption based on psychological traits and product characteristics, which makes it possible for marketers to provide customizing products and services as well as advertising messages for young consumers groups who experience high levels of FoMO when establishing their marketing strategies.

**Keywords:** bandwagon consumption; fear of missing out; luxury; necessity; consumption behavior

## 1. Introduction

Over the last few years, there has been an unusual phenomenon of collectivism relating to consumption in which consumer behavior conforms towards the pursuit of specific brands [1]. For instance, Korean women in their 20s and 30s are commonly seen holding a Starbucks cup and enjoying the ambience of a Starbucks coffee shop [2]. Similarly, the Discovery expedition or the North Face jacket, which become known as the 'teenagers' secondary school uniform', are must-have items for middle and high school students in South Korea [3,4]. The strong desire and preference for a particular brand accelerates this consumption pattern behavior [5]. This excessive consumption behavior frequently appears in eastern countries such as Korea and China [1]. Previous studies explain this mass consumption of particular brands as conformity consumption or following-up consumption [6,7]. This bandwagon consumption pattern deeply illustrates the phenomenon of collective consumption behavior which emulates the actions of others, and represents one's social status, the belonging of a particular group, and conspicuous consumption.

Bandwagon consumption refers the people's desire to purchase a commodity so they can get into 'the swim of things' or to consumers following someone they are connected with [8]. The basis of bandwagon consumption behavior is that consumers purchase products and brands not only from a utilitarian perspective but also for non-utilitarian reasons such as symbolism and increased status, while the concept is derived from conspicuous consumption [9]. Previous studies have focused on the

conceptualization and theorization of bandwagon consumption. The study of Lascu and Zinkhan [10] reveal the multiple influential factors affecting bandwagon consumption, especially the relevant tasks and situations, brand, group, and personal characteristics. The majority of recent studies on bandwagon consumption concentrate on psychological perception elements; especially, such as status signaling and association with affluent lifestyles in order to investigate what antecedents drive bandwagon consumption [9,11–13]. By examining bandwagon consumption, previous studies have identified human basic psychological traits such as social superiority and belonging to a main group.

In this context, an important question is why do consumers, in Korea, and China, conform to this excessive bandwagon consumption and become obsessed with particular brands, especially luxury brands? Is this collective consumption an intrinsic Eastern psychological assimilation of collectivism that provides stability through the pursuit of products? The typical type of bandwagon consumption which concerns higher quality, premium products, represents uniqueness and also provides a special meaning that defines status to others [10,14]. Consumers, especially from the younger generations, can conform to cost-efficient products, and as practical consumers, seek to purchase useful, economical goods that have utility and reliability. Personal or group psychological traits such as susceptibility to a reference group, and influences from celebrities, can all influence bandwagon consumption. These traits translate into a form of social superiority and feeling a sense of belonging to a main group [15–17]. Also, specific special cultures and cultural forces may affect bandwagon consumption [17,18]. For instance, people all around the world, especially the younger generation, are fanatical towards BTS (a Korean boy band) music, messages, and dance performances. Many East Asians enjoy watching Korean TV dramas and movies. Similarly to Hippie or Otaku culture, this Korean wave has drawn consumers' attention and hence many students from other countries increasingly prefer to study in Korea, and purchasing Korean pop music and other related products [18].

Despite the contribution of extant studies covering various research areas and antecedent factors, these seem somewhat limited in fully explaining excessive bandwagon consumption (e.g., the deflection of consumption behavior of a particular group). It is often considered that those who are in their late teens, 20s, and early 30s pursue a relatively rational consumption pattern as reasonable consumers [1]. But this rational consumption behavior does not entirely occur, as irrationality appears to be a part of pursuing expensive luxury goods, regardless of their price or other factors. In particular, a recent report shows that Asian international students in Korea, who regard themselves as being reasonable consumers, are particularly interested in luxury goods [19]. It may be difficult to understand this phenomenon easily, because these consumers could prefer relatively cheaper products with respect to necessity items, so this study focuses on addressing the psychological basis for the phenomenon. This aim differs from most existing research, which tends to explain the aspect of belonging to the main group, and the anxiety to deviate from the mainstream [15,20]. The phenomenon is also totally distinct from East Asia's collectivism or certain cultural factors such as Hippie, Korean wave, and Otaku trends. In order to fill this unexplored gap, this research addresses psychological traits and the Fear of Missing Out ("FoMO") principle, as methods to describe the concepts of following perceived abnormal consumption behavior, belonging to a mainstream group, and assimilating into local groups [21,22]. Apart from psychological traits, a consumer's inclination toward bandwagon consumption can vary according to the particular objects of bandwagon consumption. For instance, the tendency towards bandwagon consumption may vary across product characteristics, as consumers may show different reactions between different product categories, such as luxury and necessity products, for example.

The purpose of this paper, therefore, is to propose an explanation for bandwagon consumption based on psychological traits and product characteristics. Moreover, this paper is concerned about this phenomenon from a Korean consumer perspective of those representing the millennial generation, as opposed to any other study group internationally or otherwise. In order to examine excessive bandwagon consumption, the concept of FoMO as a psychological trait possessed by a group, is accordingly assessed. Furthermore, product characteristics (luxury/necessity) are assessed as another salient element that contributes to the formation of consumer attitudes, behaviors, and

disposition. This is important as Asian international students in Korea, for example, may have different responses towards product characteristics depending on whether they are luxury products or necessities. Hence this research combines FoMO levels and product category differentiation to explain the bandwagon consumption behavior when exploring the combinations of psychological trait and product characteristics.

## 2. Theoretical Background

### 2.1. Bandwagon Consumption Behavior

As a part of a community, individuals are prone to be affected by a group's thoughts, attitudes, and beliefs [23]. For group members this can, intentionally or otherwise, affect their product and brand choices, and hence reflect on the values upheld by that group. Being a member of a group is in itself a form of conformity, and group values and norms can thus have a substantial influence on a member's consumption behavior and decision-making when that member is very eager to be part of the mainstream [14]. This phenomenon is referred to as the 'Bandwagon Effect'.

Bandwagon consumption behavior exists when consumers buy a certain product to associate themselves within a specific group, and moreover obtain recognition from their own group [19]. This action describes the increasing popularity of a product or phenomenon that encourages more people to 'get on the bandwagon' and indicates a person's aspiration towards procuring a brand in order to associate with a particular part of the mainstream [15]. In other words, people are often inclined to imitate the consumption behavior of a respected (arguably higher) class [24]. Leivenstein, who is a pioneer of the bandwagon effect, researched consumption behavior from the perspective of social motives and social needs, and theorized three kinds of effects on material consumption: the Veblen, Snob, and Bandwagon Effects [19]. The Veblen Effect proposes that the consumer believes that price is a prestige indicator. The Snob Effect is similar, but describes the consumer as a seeker of unique value, which will therefore buy brands and products that others do not have. The Bandwagon Effect, however, describes the consumer as one that seeks a perceived social value, and thus values consumer behavior more highly than price [25,26].

Previous studies on bandwagon consumption mainly focused on brand characteristics such as popularity, quality, and the differences between competing brands [10,14]. Tsai et al. [19] proposes that product popularity among celebrities, sports stars, and other role models of influence all enhances quality perception, and moreover results in bandwagon consumption. Due to product popularity, this bandwagon consumption behavior is (re)created when consumers observe the consumption behavior of others and purchase the same brands and products—"to be one of the boys" [15]. Another perspective of recent research on bandwagon consumption were related to personal characteristics; examples of relevant personal characteristics include personal traits, external stimuli, and group characteristics such as cultural forces [27]. Recently, much research has been conducted on why consumers are so enthusiastic about purchasing luxury products [7,19,28]. This type of consumption concerning luxury goods reflects prestige and higher-class status in addition to its function and utility [29,30]. Balabanis and Kastanakis [15] significantly contribute to the bandwagon consumption argument by providing insights from the consumers' perspective. Empirical evidence suggests that consumers who are dependent on others (by having aninterdependent-self) show similarity to other consumers [15,27]. Furthermore, Parilti and Tunc [31] suggest that anxiety can also influence bandwagon consumption. Consumer anxiety theory states that personal consumption can sometimes cause anxiety in an individual, due to their perception that choices of consumption will be judged and influenced by others [32]. According to the Consumer acculturation concept, individuals in individualistic cultures tend to have an independent view of the self that emphasizes uniqueness, whereas individuals in collectivistic cultures tend to have an interdependent view of the self that emphasizes conformity [33–35]. Such a society, unlike an individualistic one, provides a consistent and overpowering need to belong to a group.

Although previous studies investigate the bandwagon consumption behavior on the basis of brand, personal, and group characteristics, little research has been conducted in the domain of a certain group or generation. Excessive bandwagon consumption of a certain group or generation, by definition, may be considered an abnormal consumption behavior, and previous studies have been limited in explaining such a phenomenon, especially in South Korea. Hence, this study explores psychological traits that are not so commonly covered, as a means to understand and explain excessive bandwagon consumption behavior.

### 2.2. The Role of FoMO on Excessive Bandwagon Consumption Behavior

### 2.2.1. The FoMO Phenomenon

Previous research on the psychological traits of bandwagon consumption has focused on why consumers purchase a specific product or brand. Moreover, research has led to psychological factors such as materialism, status-seeking predispositions, value consciousness, susceptibility of the reference group, and the need for uniqueness [13,15,36]. In particular, as in the case of the research conducted by Kastanakis and Balabanis [15], much research emphasizes the influence of psychological factors on bandwagon consumption, by investigating the self-concept (meaning the interdependent-self as opposed to the independent-self). However, these studies have some limitations in regards to explaining why consumers want to belong to a group, and moreover, why they show anxiety towards detachment from the mainstream group by abnormally purchasing a specific brand. In order to address this situation, this study adopts the conceptualization of FoMO as a psychological trait and examines bandwagon consumption behavior based on this FoMO phenomenon.

FoMO is a psychological condition where individuals want to stay continuously associated with what others are doing and hence to be free from any anxiety caused when isolated from these wishes [1,21]. Examples of FoMO behavior include browsing social networking sites and discovering oneself as not being associated with a friend's party or new topic, which can invoke fear and anxiety [21]. In order to reduce this anxiety and psychological stress, people may thus become more diligent in following others [37]. This may subsequently lead to a negative influence such as social networking site (SNS) overuse, phubbing, smartphone addition, problematic socialization, and other adverse health consequences [38–42].

### 2.2.2. FoMO and Excessive Bandwagon Consumption Behavior

The FoMO studies from a psychological angle are heavily skewed towards certain elements, such as uneasiness or apprehension towards being detached from a main group, and furthermore, describe such behavior as being a source of addiction. In particular, much research regarding FoMO focuses on the media field, an area which stimulates an emotional state, can cause obsessive use of social media platforms such as SNSs or smartphone addiction [1]. Hence, most research on FoMO is related to negative online impacts such as excessive SNS engagement and game addictions, especially among adolescents. FoMO can lead to adolescents spending much more time on online activities, thus leading to addiction [38,39,43].

Recent research undertaken has supported earlier findings that FoMO reveals an anxiety caused fear of detachment from a group, and also that online social bonding and a strong sense of belonging to the mainstream are salient factors in FoMO. One of the major findings of research on this topic is the relationship between FoMO and both online and offline behavior, as shown in Table 1.

**Table 1.** The role of the Fear of Missing Out (FoMO).

| Areas | | Role of FoMO | Author(s) |
|---|---|---|---|
| Online | SNS | Whether FoMO mediates motivation for learning and social media engagement in classes. | Alt [38] |
| | | How people with high FoMO act on internet platforms such as SNSs to increase their connections with other people | Wang et al. [44] |
| | | The relation between FOMO and increased stress associated with Facebook use among adolescents. | Beyens et al. [39] |
| | | The relationship between FOMO and problematic social network use. | Oberst et al. [45] |
| | | How FoMO influences individuals with a low/high level of basic need satisfaction when engaging with social media. | Casale and Fioravanti [40] |
| | Phubbing | The relation between FoMO and phubbing during problematic Instagram use. | Balta et al. [46] |
| | smartphone addiction | The role of FoMO on smartphone addiction. | Chotpitayasunondh and Douglas [41] |
| | | The research tried to reveal the role of FoMO as a mediator between envy and adolescent problematic smartphone use. | Wang et al. [43] |
| | Game addiction | How FoMO influences the relationship between social identity and online game addiction. | Duman and Ozkara [47] |
| Offline | Need to belong | The research looks for FoMO tendencies in adolescents with anxiety and depression. | Desjarlais and Willoughby [48] |
| | Poor life satisfaction | The research examine the relationship between FoMO and low levels of life satisfaction and general moods. | Przbylski et al. [21] |
| | Negative life impact | This research tries to look for a relation between FoMO and alcohol intake along with distracted learning/driving. | Riordan et al. [42,49] |
| | Positive social bond | How to influence FoMO regarding social interaction and social approval. | Alt and Boniel-Nissim [50] |

Previous research revealed that people who had a reduced feelings of basic satisfaction are more commonly involved in online activities to show their identities, connect with peers, and seek to belong to the mainstream [40]. Likewise, research on offline FoMO reveals an association with the need to belong and poor life satisfaction; both of these outcomes produce a negative impact [21,42,48]. Interestingly, in the Alt and Boniel-Nissim study [50], people with higher levels of FoMO tended to have increased interest in positive interactions and obtaining social approval. A common feature of online and offline FoMO found by research is that both lead people to aspire to gain a high level of perceived stability and belonging. In addition, the higher levels of FoMO people show, the greater their desire to belong to a group.

Existing studies on FoMO assert that a social state of detachment commonly prevails, especially amongst younger generations, when they deviate from an idealized form of society [22]. Consumers may instead imitate others so as to eliminate their fear of instability, anxiety, concern, and uneasiness [38]. Therefore, the stronger the desire to enter a group, the greater the bandwagon consumption behavior persists for specific brands or products such as The North Face, and Starbucks (given as examples of consumption influential products driven by unusual consumption behavior). When experiencing the risk of mainstream deviation, consumers increase their need for conformity from their surroundings. From this process, FoMO can stimulate consumers towards unreasonable consumption behavior in specific areas and exert pressure to purchase particular brands or products. That is, FoMO

represents a conceptualization of a psychological trait, in that people do not want to be excluded from the mainstream, and significantly, this psychological trait is usefully exploited via consumer behavioral research. Therefore, it is reasonable to consider FoMO as a psychological state that can tend towards unconditional conformity, and is therefore a salient factor in analyzing excessive bandwagon consumption behavior.

### 2.3. Bandwagon Consumption on FoMO Level and Product Category

From the discussion of this study on FoMO, researchers paid attention to the role of FoMO so as to examine people's consequential reactions to this emotion. Moreover, they revealed a link between the level of FoMO and behavior, and indicated that the level of FoMO has a direct relationship with the level of involvement in social media engagement, positive social interaction, and wanting social approval [50,51]. In this way, FoMO can stimulate consumption behavior by increasing the observing of others and encouraging people to follow mainstream tendencies. The level of influence could largely depend on the degree of FoMO experienced. However, it is important to conclude that FoMO is not the only influential factor in this context.

As well as the FoMO phenomenon (psychological traits), product characteristics may also influence bandwagon consumption. Product characteristics can explain why a consumer chooses a product and what purpose it is used for. Products may be classified in terms of appearance, consumer valuation, and lifestyle image, for example [12]. Moreover, products can be separated by their necessity or luxury status, based on their degree of functionality and symbolism. They may be public or private products, depending on the scope of expression available for these products. Also, they may be deemed to be hedonic or utilitarian, depending on whether they have a purpose for pleasure or practicality [52].

Generally, luxury goods as a symbolic product, place emphasis on product quality, status symbolism, and brand exclusiveness [19]. On the other hand, necessity goods are considered as a functional product and their appeal comes from a price and utilitarian perspective [53]. Lascu and Zinkhan [10] assert that with luxury goods, visibility is positively associated with the bandwagon consumption, whilst necessity and convenience goods are negatively associated. Moreover, Bearden and Etzel [54] propose a 'necessity-luxury item dimension'. The necessity-luxury item dimension compares products based on their ownership; necessities are 'possessed by everyone', whereas luxuries are products that possess a degree of exclusivity [54]. Hence much previous research on bandwagon consumption has focused on the association of luxury goods consumption with higher status and special symbolism [12,24,28]. Whilst research on this association is useful, the research has mostly been limited in explaining the concept of abnormal, excessive bandwagon consumption.

Considering Asian international students' circumstances in Korea, they can be seen to possess two kinds of identity, insomuch as they are foreigners as well as students and play a role within an unfamiliar environment [55]. Hence, the common opinion is that they are reasonable consumers with no significant income and they prefer well-functioning, practical, and price-sensitive goods, and, unlike Koreans in their late teens, 20s, and early 30s, they are likely to show a greater tendency toward focusing on necessity goods in order to lower their risk of high product usage. Based on the above discussion, bandwagon consumption is inextricably connected to product category (i.e., luxury-necessity) as well as FoMO (psychological trait), so it is relevant for this research to focus on analyzing the influences of bandwagon consumption behavior.

## 3. Hypothesis Development

### 3.1. FoMO and Bandwagon Consumption Behavior

As Asian international students have different backgrounds and cultures, they may feel that they are outsiders with a strong desire to 'belong', leading them to follow others and purchase host brands to aid assimilation, or alternatively connect to their native heritage [56]. They are generally better educated and more open minded, and as cosmopolites, are more susceptible to be influenced by

foreign cultures [57]. This means that the role of FoMO is influential for Asian international students and FoMO can play a key role in bandwagon consumption.

The collective behavior of consumers that belong to a main group may be represented by the concept of FoMO. However, it is obvious that Asian international students as consumers do not have the same level of FoMO, and that different levels of FoMO will drive them towards different behavioral outcomes [21,49]. Based on previous studies, people with higher levels of FoMO spend more time on online activities, such as social media engagements in response [38,51,58,59]. Those who have high levels of FoMO tend to drink more during large scale events, to feel a sense of belonging to a group, achieve psychological stability [49], to think of others as a part of himself/herself [60], and show positive social behaviors towards others [50]. It appears that people with higher levels of FoMO stimulate associations with the mainstream that they wish to join. Hence, it would be reasonable to divide FoMO into different levels when explaining bandwagon consumption behavior. Therefore, this study presents this following hypothesis:

**Hypothesis 1 (H1).** *Higher levels of FoMO lead to stronger bandwagon consumption behavior*.

### 3.2. Bandwagon Consumption Depending on FoMO Level and Product Category

As mentioned above, those with high FoMO levels show greater eagerness to belong to a group and demonstrate a greater need for social approval [50]. Consumers with high levels of FoMO prefer to purchase conspicuous goods to display a desirable status, show ostentation, and stay connected with their peers. Consumers purchase luxury goods not only because of their perceived high quality, but also to feed their social needs. Luxury goods are observed as images in the minds of consumers that link to high levels of price, quality, aesthetics, rarity, and extraordinariness. Moreover, they represent many non-functional associations, which reflect authority, personality, and high levels of relatedness. [29,61,62]. Conversely, it follows that for those with a low level of FoMO, Asian international students as consumers would show a smaller tendency towards bandwagon consumption than those with high levels of FoMO. This suggests that these students should act as reasonable consumers due to the limitations on their student income and an unwillingness to follow others blindly, and instead should focus on the price and value of goods. In other words, those with a low FoMO should be less influenced by peers with regards to their choices of consumption. Moreover, when they do make a purchasing decision, they pay more attention to functional demand rather than to non-functional demand.

When purchasing luxury goods, people are often likely to be influenced by factors such as price, quality, uniqueness, and the group they aspire to belong to. As a result of high levels of FoMO, people tend to remain in the mainstream. As such, this can lead to consumer adherence to an associated, main group and lead to strong bandwagon consumption behaviors. On the other hands, financial considerations tend towards reduced conformity responses compared to luxury, since consumers do not prioritize status and mainstream belonging so highly under these circumstances [10]. This would lead people to be reasonable consumers that do not follow others blindly, and these consumers show a tendency towards a low FoMO. From the above discussion, it can be inferred that Asian international students, as consumers, reveal different bandwagon consumption behavior based on product categories. Based on FoMO levels and product category, this study could infer that Asian international students exhibit various bandwagon consumption behaviors, and present the following hypotheses, research model (Figure 1):

**Hypothesis 2 (H2).** *Groups with higher levels of FoMO have stronger bandwagon consumption on luxury goods than on necessary goods*.

**Hypothesis 3 (H3).** *Groups with lower levels of FoMO have stronger bandwagon consumption on necessary goods than on luxury goods*.

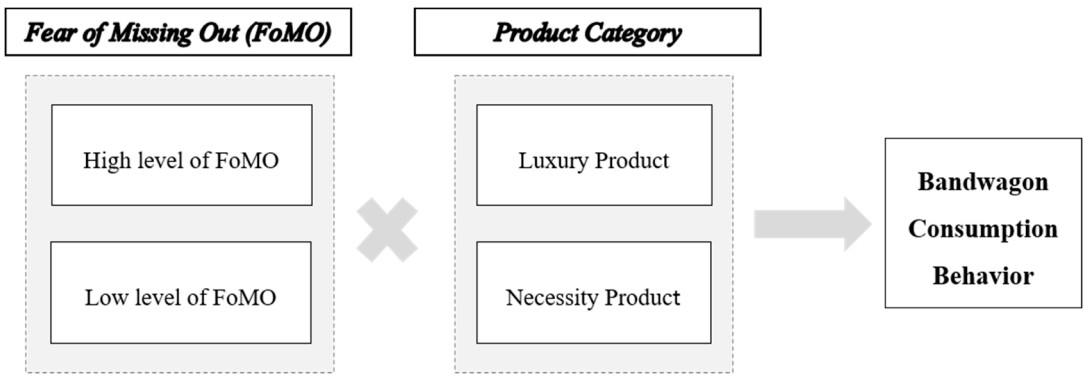

**Figure 1.** Research Model.

## 4. Methodology

### 4.1. Participants and Procedure

In order to study the consumer bandwagon consumption behavior based on different levels of FoMO and product categories (luxury/necessity), we conducted a survey of visiting Asian international students in Seoul. According to data released by the National Institute for international education, 67% of Korean international students are from China and Vietnam (74% of Asian international students), with most between 18 and 34 years of age [63]. They are emerging as a new consumer class of foreign consumers with increasingly affluent wealth [64]. Firstly, this research adopted the concept of FoMO, which is divided into a high level of FoMO and a low level of FoMO. This makes it possible to assess how consumers show their inclinations towards bandwagon consumption behaviors depending on their levels of FoMO. Secondly, we needed to select the representative brands of luxury and necessity for Asian international students. For luxury, "Sulwhasoo" and "Whoo", which are highly premium, luxury cosmetic brands among Chinese tourists [65], were selected. In particular, Sulwhasoo is perceived to be the most premium Korean brand, containing highly valued Korean herbal medicinal ingredients. It has been the most successful premium cosmetics brand in Korea since 2012 and has surpassed the sales of other international brands such as Chanel, Estee Lauder, and SK-II [66]. The prices of these skincare products range from US$150 to US$700 per 50 mL. Compared to other worldwide luxury brands such as Estee Lauder or Lancome, the prices for these Korean brands are similar. Because the brand images and prices of Korean luxury cosmetic brands are competitive with international cosmetics brands, the above Korean brands would be regarded as being luxurious brands for university students. For the necessity category, this research selected the following brands: "Ryeo (shampoo)" and "Aekyung 2080 (toothpaste)", which range from US$2 to US$8 and are much cheaper than the above luxury brands.

### 4.2. Sampling

In order to determine whether each survey question could be easily understood, a preliminary questionnaire was distributed to approximately 10% of the final sample size (25 Asian international students) who purchased and used the above luxury and necessity brands. The offline survey required Asian international students to answer each question, and this process using quota sampling methodology, was carried out over a 2 week period from 3 October to 16 October in 2019. Following this, a total of 320 questionnaires were handed out and 255 were returned. This research accepted 218 of these returned questionnaires for analysis, but excluded 37 questionnaires as being invalid, after deciding the answers given did not entirely relate to the questions posed.

### 4.3. Measures

FoMO. For the analysis of this research, all variables were referenced from previous literature. FoMO was measured using 3 items from the study of Przybylski et al. [21]. They are: "I get worried

when I find out my friends are having fun without me.", "I get anxious when I don't know the brands popular among my friends.", and "It is important that I continue to keep in touch with what my friends are doing." The respondents answered each questionnaire to their level of agreement with each question based on a 5-point Likert scale (1 = not at all true of me, 5 = extremely true of me). The responses were averaged to form a scale (M = 3.523, SD = 0.874) and indicated adequate reliability (Cronbach's α = 0.878).

Bandwagon Consumption Behavior (Luxury/Necessity). Each respondent addressed any bandwagon consumption of luxury brands and necessity brands accordingly. Bandwagon consumption was examined using a method from a previous study by Kastanakis and Balabanis [15], which consists of 3 items with a 5-point Likert scale (1 = not at all true of me, 5 = extremely true of me); "How likely is it that you would purchase/use products worn by most people?", "How likely is it that you would purchase/use popular products that everyone would approve of?" and "How likely is it that you would purchase/use products recognized by many people?" The responses were averaged to form a scale (Luxury: M = 3.529, SD = 0.852; Necessity: M = 2.824, SD = 0.811) and indicated adequate reliability (Luxury: Cronbach's α = 0.879, Necessity: Cronbach's α = 0.849).

## 5. Findings

### 5.1. Sample Characteristics

The sample for this analysis consisted of 48.6% male and 51.4% female respondents. With regards to nationality, Chinese students represented the largest sample contingency (64.7%), followed by Vietnamese students (19.3%). This ratio was similar to the released data of the Institute of International Education in 2018, which indicated that Chinese and Vietnamese students account for 74% of Asian international students in South Korea. Of the other nationalities used, 4.1% were from Japan, 3.2% were from Mongolia, and 8.7% were from other Asian countries. Details are shown in Table 2.

**Table 2.** Sample characteristics (n = 218).

| Item | Characteristics | Frequency | Ratio |
|---|---|---|---|
| Gender | Male | 106 | 48.6 |
| | Female | 112 | 51.4 |
| | Total | 218 | 100.0 |
| Nationality | China | 141 | 64.7 |
| | Vietnam | 42 | 19.3 |
| | Japan | 9 | 4.1 |
| | Mongolia | 7 | 3.2 |
| | Others | 19 | 8.7 |
| | Total | 218 | 100.0 |

### 5.2. Data Analysis

This research aimed to study and verify the effect of bandwagon consumption for levels of FoMO and product category. The main concept of this research was an two-way ANOVA analysis using 2 (High FoMO/Low FoMO) by 2 (luxury/necessity). To establish more detail on how the level of FoMO with the product category affects bandwagon consumption, quantitative data were collected and FoMO was divided into 2 levels (high FoMO, low FoMO) at the median level. Using SPSS 23.0 software, independent T-test, one-way ANOVA, and two-way ANOVA with estimated marginal means were calculated.

### 5.3. Empirical Results

Before analyzing the bandwagon consumption associated with the FoMO level and product category using ANOVA, this research conducted a planned-contrast analysis using an independent

T-test based on FoMO level so as to reveal each group mean, each group difference, and each F-value. The result showed each group's difference, and the details are shown in Table 3.

**Table 3.** Bandwagon Consumption depending on the level of FoMO: Independent T-test.

| Luxury | | | | Necessity | | | |
|---|---|---|---|---|---|---|---|
| **FoMO Level** | **Mean(SD)** | **Difference** | **t-Value** | **FoMO Level** | **Mean(SD)** | **Difference** | **t-Value** |
| High (N = 118) | 4.18 (0.43) | 1.427 | 22.456 *** | High (N = 118) | 2.17 (0.38) | −1.421 | −26.586 *** |
| Low (N = 100) | 2.76 (0.51) | | | Low (N = 100) | 3.59 (0.41) | | |

Levene's test (Luxury: F = 0.998, Sig. = 0.319, Necessity: F = 0.144, Sig. = 0.705), *** $p < 0.001$.

The above analysis indicates the mean, mean difference, t-value, and Levene's test. Based on product category, the mean difference between high FoMO and low FoMO shows 1.427 for luxury and −1.421 for necessity a 99.9% confidence level (t-value of luxury: 22.456, t-value of necessity: −26.586). This means that between high FoMO and low FoMO, there are significant differences between luxury brands and necessity brands. Nevertheless, in the case of necessity brands, the result for bandwagon consumption showed that the group with a low level of FoMO had more resistance to this form of consumption than that the group with a high level of FoMO. That is, a higher level of FoMO has influence on bandwagon consumption only for luxury brands. So, H1 was not supported (Luxury was supported but necessity was not supported).

In order to achieve this paper's objective, this study firstly adopted an ANOVA using SPSS 23.0 software. Table 4 shows the result of the bandwagon consumption effect of the FoMO level (main effect), product category (main effect), FoMO level * product category (interaction effect) with the mean, mean difference, and significance using one-way ANOVA and two-way ANOVA.

**Table 4.** Bandwagon Consumption depending on level of FoMO and product category (ANOVA).

| Dependent Variable: Bandwagon Consumption | | | | | |
|---|---|---|---|---|---|
| **FoMO Level** | **Product Category** | | **Mean Difference** | **Std. Error Difference** | **Sig.** |
| | **Luxury** | **Necessity** | | | |
| High FoMO | 4.18(0.43) | 2.17(0.38) | 2.011 | 0.056 | 0.000 |
| Low FoMO | 2.76(0.51) | 3.59(0.41) | −0.837 | 0.066 | 0.000 |
| Test of Between-subject effects | Source | Type III sum of squares | Mean square | F | Sig. |
| | FoMO level (A) | 0.001 | 0.001 | 0.005 | 0.943 |
| | Product category (B) | 37.342 | 37.342 | 200.111 | 0.000 |
| | A * B | 219.515 | 219.515 | 1176.347 | 0.000 |

* R squared = 0.772 (Adjusted R Squared = 0.771).

First, with a high level of FoMO, luxury goods show a stronger bandwagon consumption pattern, which was 2.011 of mean difference with a 0.000 significance level. On the other hand, with a lower level of FoMO, necessity brands showed a stronger bandwagon consumption level, which was −0.837 of mean difference with a 99.9% significance level. Next, the FoMO level of the main effect showed no significant difference for bandwagon consumption. This means the level of FoMO would not lead to bandwagon consumption (H1 is therefore not supported). Second, the product category's impact on bandwagon consumption showed a major effect with a 99.9% significance level. Next, looking through the interaction effect, interaction between the FoMO level and product category indicated significant results with a significance level of 99%. Based on the above analysis, H2 and H3 were supported.

The above mentioned, the FoMO level and product category showed a significant correlation with a 99% confidence level, which means that they have interaction effects. Based on optional analysis of ANOVA, more details of the estimated marginal means on the bandwagon effect are shown as Table 5.

**Table 5.** Estimated Marginal Means (Level of FoMO * Product category).

| | | Dependent Variable: Bandwagon Consumption | | |
|---|---|---|---|---|
| **FoMO** | | **Product Category** | | FoMO Level * Product category interaction |
| | | **Luxury** | **Necessity** | |
| High | Mean | 4.18 | 2.17 | |
| | (SD) | 0.040 | 0.040 | |
| Low | Mean | 2.76 | 3.59 | |
| | (SD) | 0.043 | 0.043 | |

## 5.4. Discussion

Chinese, Vietnamese, and other Asian international students do subscribe to the bandwagon consumption principle, mainly because they have similar values to those of Koreans (such as Confucian values) and are more easily influenced by the Korean life patterns and ways of thinking [67]. Importantly, Asian international students are influenced by the collectivistic culture, and will exhibit group-oriented consumption inclinations more and prefer bandwagon brands [19]. Thus, based on the above results, this research confirms how Asian international students in Korea reacted to bandwagon consumption behavior in regards to the FoMO phenomenon and product category. Below are the characteristics of each group:

① **High FoMO and Luxury:** Mean and estimated marginal means for high FoMO and luxury are the highest results and indicate a significant influence on bandwagon consumption compared to other consumption behaviors. Higher levels of FoMO would stimulate bandwagon consumption through increased aspirations to join 'the mainstream in-group' and purchasing luxury brands representing exclusiveness and a status symbol makes it possible to feel a sense of belonging to a reference group with similar goods [19]. Also, Asian international students, especially from China and Vietnam, have mostly grown up in a relatively wealthy environment, so they have enough resources to enjoy life and purchase popular products with the financial support of their parents [19]. Based on the above consumers characteristics, Asian international students have exhibited the strongest bandwagon consumption behavior compared to other groups.

② **High FoMO and Necessity:** Mean and estimated marginal means for this group were the third-highest results. In spite of a higher level of FoMO, necessity brands did not show strong bandwagon consumption behavior. This means that Asian international students would not be induced to purchase necessity brands since necessity brands do not stimulate one's psychological stability to stay connected with peers or persons of the main group they want to be associated with. Necessity brands are considered as functional, utilitarian products. In order for Asian international students with a higher level of FoMO to satisfy their needs for belonging to aspirational groups, they prove themselves by purchasing symbolic brands or exhibiting luxury brands. That is why people in this group shows weaker bandwagon consumption behavior than those who are in the group with high FoMO and luxury brands.

③ **Low FoMO and Luxury:** The mean and estimated marginal means were the lowest for the results of this group. Those with a low FoMO are less willing to belong to a desired and aspired group. Under this psychological state, they do not need to show their status symbol, follow others blindly, or consume goods conspicuously. These people cannot stimulate luxury consumption behavior, since they need to pay more attention to the price, quality, and functional demands of products. Consumers in this group tend to show less conformity consumption behavior. Hence,

the needs of purchasing luxury brands would be relatively weaker and lead to the lowest mean of bandwagon consumption behavior compared to all the other consumption behaviors examined in this research.

④ **Low FoMO and Necessity:** The mean and estimated marginal means were the second-highest and this had a significant influence on bandwagon consumption for this group. Contrasted with the normal assumption that the tendency towards bandwagon consumption is the lowest for this group, Asian international students in this group show the second-highest bandwagon consumption behavioral trend. This indicates those who have a low FoMO are very practical, price-sensitive, and have a risk-free disposition in order to reduce failure probability [68]. Regardless of psychological traits such as belonging to a main group and anxiety of deviation, they, as foreigners and students, seek to be reasonable, risk-aversive consumers within an unfamiliar environment [55,68].

Based on the above discussion, there are significant differences toward bandwagon consumption based on an individual's level of FoMO and product categories. As expected, the combination of high FoMO and luxury is the highest combination found. This means that many Asian international students in Korea want to be part of the main group in Korea, so under high FoMO, their consumption of luxury products is higher than that of any other group. The next most common category is low FoMO and necessity, followed by high FoMO and necessity, and finally low FoMO and luxury. The second most common category suggests that Asian international students in Korea only purchase necessities for practical purposes.

## 6. Conclusions

Through the verification of empirical tests, the implications of this study are summarized below. Firstly, in contrast with previous studies of bandwagon consumption that investigate issues such as social norms and brand characteristics, this study has a psychological basis, as it focuses on the FoMO phenomenon and utilizes it to explain bandwagon consumption relating to Asian international students in South Korea. This is a phenomenon that has been largely unexplained in existing studies. This study investigated the importance of FoMO in consumer behavior in relation to belonging to a desirable group and has presented a case which proposes that consumers display significantly different tendencies depending on their level of FoMO. Hence, the greater the FoMO a consumer feels, the greater their tendency is towards bandwagon consumption. This proposed principle offers an advanced method of research through psychological traits in the field of consumer behavior and marketing.

Secondly, this study examines bandwagon consumption through a mixture of varying degrees of FoMO and product characteristics. The findings show that there are significant differences between FoMO (high/low) and product characteristics (luxury/necessity). Those persons with a high level of FoMO tend more towards purchasing luxury rather than necessary goods; on the other hand, those who possess low levels of FoMO tend to purchase more necessary goods over luxury goods. Based on an empirical test on bandwagon consumption, a high level of FoMO with luxury goods was found to be the most common bandwagon consumption behavior among the respondents. Next was individuals that were low in FoMo towards necessary goods, followed by high FoMO towards necessary goods, and last was low in FoMO towards luxury goods. From the perspective of Asian international students, individuals high in FoMO purchase luxury goods due to their symbolic value and to improve their status, hence indicating their aspiration to belong in a desirable main group. Conversely, individuals low in FoMO purchase necessity goods predominately due to their functional value and to reduce excessive product usage. In conclusion, for consumers with high FoMO towards luxury goods, the dominant norm or standards of a social group to which an individual belongs may drive them to a new style or products in parallel with other group members. Under these circumstances, it may be difficult for an individual to make a purchasing decision independent of the influence of others. So, consumers with this tendency will be more likely to perform more follow-up consumption than those who do not have anxiety towards missing out from social group membership.

Thirdly, this study strongly supports the acculturation theory principle suggesting that consumers adapt to the society in which they live, acquire the lifestyle of those around them, and thus assimilate into other groups, even across borders. In this study, the survey respondents used are Asian international students residing in Korea, and their ages vary between late teens and early thirties. Even though their main purpose for staying in Korea is for studying, they are Korean-like consumers with adopted Korean consumption tendencies, and are especially familiar with the ideology of collectivism.

Based on the results of this paper's conducted research and analysis, this paper presents certain managerial implications as follows: Firstly, practitioners would greatly benefit by using the conceptualization of FoMO for marketing mix purposes and advertising strategies; especially for luxury brands, as FoMO is a useful characteristic to stimulate bandwagon consumption behavior, emphasizing status, symbolism, and the feeling of belonging.

Secondly, this research indicates the importance of the consumption trends of Asian international students in Korea, who are mostly from China, Vietnam, and other Asian countries, and belong to the millennial generation. They have a tendency of assimilating culturally, both in groups and within their recognized mainstream identifications. They pay attention to others' behavior, not least because they relate to Confucian culture and collectivism. Marketers need to apply this psychological trait to stimulate consumers to be part of a "main-group". This provides another marketing opportunity for specific groups or generations, especially in Korea, China, and other Eastern Asian countries. Xiahongshu, which is a social media and e-commerce platform in China, provides customizing products as well as services as well as advertising messages for particular groups. It would be strategic to present advertisement messages that are effective for young consumers who have high levels of FoMO [1]. For marketers, it is necessary to establish marketing strategies, and conduct marketing activities directed towards those consumers.

## 7. Limitations and Future Research Directions

Even though this study has made the case discussed above, there are several limitations. Firstly, this study focuses on FoMO as a psychological trait to conduct an evaluation of bandwagon consumption. FoMO is not the only psychological trait likely to explain excessive bandwagon consumption. Future research is necessary to identify other germane psychological traits to investigate which ones are the most influential, when considering culture, domesticity, gender and age, for example.

Secondly, in regards to sample bias, the data in this research were collected from a sample of Asian international students staying in Korea, of which over 50% were from China. Chinese students may be more influenced by collectivist culture, exhibit group-oriented consumption, and prefer bandwagon brands more than their European or American counterparts, and hence the culture factor must be kept in mind. Thus, it would be useful for future research to investigate the effect of culture differences on bandwagon consumption, especially on those consumers who are from countries influenced by longer-held traditions, such as England, France, and the USA.

Lastly, this research did not discuss the effect of bandwagon consumption from the perspective of social comparison. People tend to compare themselves with others [69]. This would influence psychological motivation of consumers towards bandwagon consumption. For future research, it could be meaningful to discuss consumption behavior using the social comparison theory.

In spite of the above limitations, this research contributes to the understanding of excessive bandwagon consumption based on a given psychological trait and product characteristics and, to marketers and practitioners, gives valuable insight into more fully understanding consumers' consumption behavior.

**Author Contributions:** I.K., and I.M. contributed significantly to the conception of the study, data collection, analysis, and to preparing the manuscript. All authors have read and agreed to the published version of the manuscript.

**Funding:** This research received no external funding.

**Conflicts of Interest:** The authors declare no conflict of interest.

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
