# Peer review of "A Study on Bandwagon Consumption Behavior Based on Fear of Missing Out and Product Characteristics"

_sustainability, doi:10.3390/su12062441_

Round 1

Reviewer 1 Report

Report Review

This paper presents an empirical study on bandwagon consumption behaviour among international Asian students, the authors conducting a well-founded research, based on a consistent theoretical background. The topic approached from a psychological perspective is interesting and may generate future studies in this direction of bandwagon consumption behavior.

However, some small changes and recommendations are required. 

  1. In the abstract, it should be mentioned and explained much more clearly why this study is practically necessary, especially since there are implications in establishing sustainable marketing strategies.
  2. At keywords it would be advisable to add the term consumption behavior or, even better, bandwagon consumption behavior.
  3. In section 1. Introduction, if „the purpose of this paper is to propose an explanation for bandwagon consumption based on psychological traits and product characteristics”, then it is necessary to specify better the characteristics of the product, as clearly specified the product category : luxury/ necessity.
  4. In sections 2.2.1. FoMO Phenomenon and 2.2.2. FoMO and Excessive Bandwagon Consumption Behavior, the term SNS is used, “SNS overuse” and “SNS engagement”, it refers to SMS? Must be explained.
  5. Also in the table 1 Role of Fear of Missing Out (FoMO) appears this term SNS and “Phubbing” instead of “Pubbing”.
  6. In section 4.3 Measures it would be advisable to present a table with conceptual definition of variables that include the model factor, indicator observed and measurement. It would give greater clarity and consistency to the presentation.
  7. In section 6. Conclusion, some modification are necessary:

“Firstly, in contrast with previous studies of bandwagon consumption that investigate issues such as social norms and brand characteristics, this study has a psychological basis, insomuch as it focuses on the FoMO phenomenon…”, in so much.

“A phenomenon that has been largely unexplained in existing studies encountered in my experience.” I think this sentence should be expressed otherwise, without having a personal touch.

“This proposed principle offers an advanced method of research through psychological trait on the realm of consumer behavior and marketing.”, much better “field of consumer behavior and marketing.”

  1. Also in this section 6, with managerial implications, the role of FoMO conceptualization in the marketing mix and the advertising strategy should be clearly explained, all the more so as it is noted at the end of the paragraph and “other marketing opportunities”. What would those be?
  2. In the abstract of the paper it is mentioned “From the finding, this study might present implications for understanding excessive bandwagon consumption based on psychological trait and product characteristic and for establishing sustainable marketing strategies.”, again this idea should be developed to managerial implications, at the end of the section 6. Conclusion.
        1.  

Author Response

General Comments: This paper presents an empirical study on bandwagon consumption behaviour among international Asian students, the authors conducting a well-founded research, based on a consistent theoretical background. The topic approached from a psychological perspective is interesting and may generate future studies in this direction of bandwagon consumption behavior.

Response: Authors are grateful to the reviewer for the positive and encouraging comments.

Comment 1: In the abstract, it should be mentioned and explained much more clearly why this study is practically necessary, especially since there are implications in establishing sustainable marketing strategies.

Response: We modified the abstract of this paper as suggested. The purpose of this study is to confirm the influence of excessive bandwagon consumption behavior based on FoMO levels and product category. From this, this study would present implications for marketers, which provide customizing products and services as well as advertising message for consumers of young group who are in high levels of FoMO. So, we changed the part of  “establishing sustainable marketing strategies” into “establishing marketing strategies” (Please see page 1, 13).

Comment 2: At keywords it would be advisable to add the term consumption behavior or, even better, bandwagon consumption behavior.

Response: As suggested, we inserted “consumption behavior” at keywords.

Comment 3: In section 1. Introduction, if „the purpose of this paper is to propose an explanation for bandwagon consumption based on psychological traits and product characteristics”, then it is necessary to specify better the characteristics of the product, as clearly specified the product category : luxury/ necessity.

Response: As suggested, we clearly specified the product characteristic (luxury / necessity) (Please see page 2)

Comment 4: In sections 2.2.1. FoMO Phenomenon and 2.2.2. FoMO and Excessive Bandwagon Consumption Behavior, the term SNS is used, “SNS overuse” and “SNS engagement”, it refers to SMS? Must be explained.

Response: As suggested, we add FoMO explanation on excessive social media use, SNS. (Please see page 4).

Comment 5: Also in the table 1 Role of Fear of Missing Out (FoMO) appears this term SNS and “Phubbing” instead of “Pubbing”.

Response: As suggested, we changed “pubbing”(in sections 2.2.1. FoMO phenomenon) into “phubbing”. (Please see page 4).

Comment 6: In section 4.3 Measures it would be advisable to present a table with conceptual definition of variables that include the model factor, indicator observed and measurement. It would give greater clarity and consistency to the presentation.

Response: When we submitted our manuscript, we considered it several times. But we expressed that way since this study did not have many variables.

Comment 7: In section 6. Conclusion, some modification are necessary:

“Firstly, in contrast with previous studies of bandwagon consumption that investigate issues such as social norms and brand characteristics, this study has a psychological basis, insomuch as it focuses on the FoMO phenomenon…”, in so much.

“A phenomenon that has been largely unexplained in existing studies encountered in my experience.” I think this sentence should be expressed otherwise, without having a personal touch.

“This proposed principle offers an advanced method of research through psychological trait on the realm of consumer behavior and marketing.”, much better “field of consumer behavior and marketing.”

Response: As suggested, all above sentences are modified. (Please see page 12)

Comments 8: Also in this section 6, with managerial implications, the role of FoMO conceptualization in the marketing mix and the advertising strategy should be clearly explained, all the more so as it is noted at the end of the paragraph and “other marketing opportunities”. What would those be?

Response: We added Xiahongshu example of marketing mix and advertising strategy the part of managerial implication as below

(Xiahongshu, which is a social media and e-commerce platform in China, provides customizing products, services as well as advertising messages for particular groups. It would be strategic to present advertisement messages that is effective for young consumers who are in high levels of FoMO. For marketers, it is necessary to establish marketing strategies, and conduct marketing activities for those consumers.)

Comments 9: In the abstract of the paper it is mentioned “From the finding, this study might present implications for understanding excessive bandwagon consumption based on psychological trait and product characteristic and for establishing sustainable marketing strategies.”, again this idea should be developed to managerial implications, at the end of the section 6. Conclusion.

Response: As suggested, based on response of comment 1, we developed the description of conclusion with several references. (Please see page 13).

Response of comment 1: The purpose of this study is to confirm the influence of excessive bandwagon consumption behavior based on FoMO levels and product category. From this, this study would present implications for marketers, which provide customizing products and services as well as advertising message for consumers of young group who are in high levels of FoMO. So, we changed the part of  “establishing sustainable marketing strategies” into “establishing marketing strategies” (Please see page 1, 13).

Reviewer 2 Report

The idea of the study is interesting, but I think it needs some improvements before publishing.

Please clarify the rationale of the hypotheses 2 and 3.

Research design should be renamed “Participants and Procedure”

Please clarify the dependent variable of the study. In the Figure 1, research model, is not clear how the variable “Bandwagon Consumption” was measured.

The section “Sample characteristics! Should be revised according to the new suggestions.

Please include a new section “data analysis” to explain how the data were analysed.

Please clarify and justify how the FoMO levels were obtained.

However, the current approach used to analyse the data does not allow to test the hypothesis of the study. I think that the Authors should consider the opportunity to use a SEM methodology to test the suggested research model. However, the main limitation of this study is the absence of a mediator variable. Could be interesting to explore the mediating role of social comparison in the relationship between FoMO and Bandwagon.

Author Response

General Comments: The idea of the study is interesting, but I think it needs some improvements before publishing.

Response: Authors are grateful to the reviewer for the positive and encouraging comments.

Comment 1: Please clarify the rationale of the hypotheses 2 and 3.

Response: As suggested, we added some rationale in order to clarify the hypotheses 2 and 3 (Please see 7).

Comment 2: Research design should be renamed “Participants and Procedure”

Response: As suggested, we renamed into “Participants and Procedure”. (Please see page 8).

Comment 3: Please clarify the dependent variable of the study. In the Figure 1, research model, is not clear how the variable “Bandwagon Consumption” was measured.

Response: Figure 1 is revised as suggested (Please see page 7).

Comment 4: The section “Sample characteristics! Should be revised according to the new suggestions.

Response: As suggested, the table name is changed into “Sample characteristics” (Please see page 9).

Comment 5: Please include a new section “data analysis” to explain how the data were analysed.

Response: As suggested, we insert 5.2 Data analysis in order to show how the data were analysed. (Please see page 9).

Comment 6: Please clarify and justify how the FoMO levels were obtained.

Response: From previous literature (Przbylski et al., 2013), FoMO levels were assessed. This research adjusted average dichotomizing method  (Aiken et al., 1991) and based on this, we divided the group into high & low FoMO levels. Using ANOVA method, we drew the value of FoMO and with other variables. (Please see page 8, 9).

Comment 7: However, the current approach used to analyse the data does not allow to test the hypothesis of the study. I think that the Authors should consider the opportunity to use a SEM methodology to test the suggested research model. However, the main limitation of this study is the absence of a mediator variable. Could be interesting to explore the mediating role of social comparison in the relationship between FoMO and Bandwagon.

Response: We really appreciated your comment on this and this would be our future research subject. The purpose of this study is to confirm the influence of excessive bandwagon consumption behavior based on FoMO levels and product category. Using ANOVA analysis, we tried to show group difference behavior (high FoMO & luxury, high FoMO & necessity, low FoMO & luxury, low FoMO & necessity).

Reviewer 3 Report

The reviewed paper is of scientific nature. The subject area discussed in the paper should be considered interesting. The structure of the paper is clear. The value of the paper results from combination of literature studies with the results of Authors' own research.

However, deliberations conducted in the paper need to be expanded. Therefore, it is specifically recommended to:

- focus considerations conducted in the paper more on the category of sustainable development and sustainable marketing strategies,

- develop the description of conclusion and refer to literature in this part of article.

Author Response

General Comments: The reviewed paper is of scientific nature. The subject area discussed in the paper should be considered interesting. The structure of the paper is clear. The value of the paper results from combination of literature studies with the results of Authors' own research.

Response: Authors are grateful to the reviewer for the positive and encouraging comments.

Comment 1: focus considerations conducted in the paper more on the category of sustainable development and sustainable marketing strategies,

Response: The purpose of this study is to confirm the influence of excessive bandwagon consumption behavior based on FoMO levels and product category. From this, this study would present implications for marketers, which provide customizing products and services as well as advertising message for consumers of young group who are in high levels of FoMO. So, we changed the part of  “establishing sustainable marketing strategies” into “establishing marketing strategies” (Please see page 1, 13).

Comment 2: develop the description of conclusion and refer to literature in this part of article.

Response: As suggested, based on comment 1, we developed the description of conclusion with several references. (Please see page 13).

Xiahongshu, which is a social media and e-commerce platform in China, provides customizing products, services as well as advertising messages for particular groups. It would be strategic to present advertisement messages that is effective for young consumers who are in high levels of FoMO. For marketers, it is necessary to establish marketing strategies, and conduct marketing activities.

Round 2

Reviewer 2 Report

The Authors improved the previous version of the manuscript. The manuscript is now acceptable for publication after minor revisions.

I think that the Authors should consider the following suggestions:

Table 1 is unusual to list the results of previous studies in similar way. I suggest discussing the studies in the text.

Page 9 – Authors in the text introduced the ANOVA, but the results showed in the table 3 are related to t-test. Please clarify this issue.

Page 10 – Please clarify which kind of ANOVA statistical method has been used. I noted a difference between the section 5.2 and the section 5.3.

In the limitation of the study, Authors should also discuss the effect of social comparison.

Author Response

General Comments: The Authors improved the previous version of the manuscript. The manuscript is now acceptable for publication after minor revisions.

Response: Authors are grateful to the reviewer for the positive and encouraging comments.

Comment 1: Table 1 is unusual to list the results of previous studies in similar way. I suggest discussing the studies in the text.

Response: As suggested, we modified the table 1 (Please see page 4, 5).

Comment 2: Page 9 – Authors in the text introduced the ANOVA, but the results showed in the table 3 are related to t-test. Please clarify this issue.

Response: As suggested, we inserted “independent T-test” in the first paragraph of part 5.3 empirical results. Before making the ANOVA analysis, we wanted to show the each group differences (mean, standard deviation and difference). So we indicated those analysis in the first, second paragraph of part 5.3 empirical results (Please see page 9).

Comment 3: Page 10 – Please clarify which kind of ANOVA statistical method has been used. I noted a difference between the section 5.2 and the section 5.3.

Response: As suggested, we clearly specified one-way ANOVA, two-way ANOVA in the paragraph (Please see page 10). The difference between the section 5.2 and the section 5.3 is that the section 5.2 explained which statistical method adopted this research and the section 5.3 indicated that what the data meant based on each statistical method (Please see page 9, 10).

Comment 4: In the limitation of the study, Authors should also discuss the effect of social comparison.

Response: As suggested, we added the effect of social comparison in the limitation of the study (Please see page 14).